# Microwave-induced conductance replicas in hybrid Josephson junctions without Floquet–Andreev states

Daniel Z. Haxell[1], Marco Coraiola[1], Deividas Sabonis[1], Manuel Hinderling[1], Sofieke C. ten Kate[1], Erik Cheah [2], Filip Krizek[1,2], Rüdiger Schott [2], Werner Wegscheider [2], Wolfgang Belzig [3], Juan Carlos Cuevas [3,4] & Fabrizio Nichele [1] ✉

Light–matter coupling allows control and engineering of complex quantum states. Here we investigate a hybrid superconducting–semiconducting Josephson junction subject to microwave irradiation by means of tunnelling spectroscopy of the Andreev bound state spectrum and measurements of the current–phase relation. For increasing microwave power, discrete levels in the tunnelling conductance develop into a series of equally spaced replicas, while the current–phase relation changes amplitude and skewness, and develops dips. Quantitative analysis of our results indicates that conductance replicas originate from photon assisted tunnelling of quasiparticles into Andreev bound states through the tunnelling barrier. Despite strong qualitative similarities with proposed signatures of Floquet-Andreev states, our study rules out this scenario. The distortion of the current–phase relation is explained by the interaction of Andreev bound states with microwave photons, including a non-equilibrium Andreev bound state occupation. The techniques outlined here establish a baseline to study light–matter coupling in hybrid nanostructures and distinguish photon assisted tunnelling from Floquet-Andreev states in mesoscopic devices.

Non-equilibrium quantum systems can be designed and manipulated using light–matter interactions. In condensed matter, spatially and temporally cyclic Hamiltonians are expected to generate energy-periodic Floquet states, with properties inaccessible at thermal equilibrium[1–9]. Together with realisation of Floquet states, periodic electromagnetic fields have a strong influence on the transport properties of nanostructures. A prominent example is a superconducting weak link with dimensions smaller than the quasiparticle inelastic mean free path and subject to a homogeneous electric field due to microwave (MW) irradiation. In this case, an AC bias develops across the weak link and hence photon-assisted tunnelling (PAT) occurs[10–12]

where quasiparticles traverse the junction by absorption or emission of photons. The PAT mechanism has successfully described the current–voltage characteristics of superconducting weak links, including Dayem bridges[13], atomic-sized point contacts[14,15] and scanning probes[16–18].

Hybrid Josephson junctions (JJs) consist of a normal semiconducting material (N) confined between two superconductors (S), where electron-hole reflection at the S–N interfaces leads to Andreev bound states (ABSs) at energies below the superconducting gap[19, 20]. Microwave irradiation of hybrid JJs has been used to study spin–orbit interaction[21–24], drive coherent transitions between ABSs[25–32] and

[1]IBM Research Europe—Zurich, 8803 Rüschlikon, Switzerland. [2]Laboratory for Solid State Physics, ETH Zürich, 8093 Zürich, Switzerland. [3]Fachbereich Physik, Universität Konstanz, D-78457 Konstanz, Germany. [4]Departamento de Física Teórica de la Materia Condensada and Condensed Matter Physics Center (IFIMAC), Universidad Autónoma de Madrid, E-28049 Madrid, Spain. ✉e-mail: fni@zurich.ibm.com

investigate PAT[33,34]. Hybrid JJs also constitute a promising platform to realise non-equilibrium quantum systems induced by light–matter interaction[35–39], particularly due to their voltage-tunable properties and their compatibility with superconducting circuitry.

Exposing JJs to MW irradiation is prone to experimental complexities. These devices are typically connected to large bonding pads —that act as antennas—and are placed within metallic enclosures—that act as cavities. Depending on device geometry, materials involved, and properties of the MW field (such as intensity, polarisation and spot size), complex electromagnetic field distributions and non-uniform coupling strength of the MW radiation to the system arise, leading to multiple phenomena. Further intricacies occurr in three-terminal devices, in which one terminal is used to probe the local density of states (DOS) in the JJ via an additional superconducting weak link[40–42]. In this case, the MW field could couple both to the JJ, enabling modifications to its DOS, and to the probe, affecting the transport processes between the probe and the junction. Moreover, a time-dependent drive of the JJ leads was shown to result, to some extent, in a time-dependent potential at the weak link[43]. This plethora of phenomena makes it challenging to differentiate the microscopic mechanisms governing the interaction between the MW radiation and the device, and motivates the search for unambiguous experimental signatures of strong light–matter coupling.

We consider the specific case of a hybrid JJ hosting discrete ABSs, which are probed by tunnelling spectroscopy using a superconducting weak link. Andreev bound states with energy $E_A$ are measured as shown by the schematic in the top panel of Fig. 1a. A current (yellow) flows between the superconducting probe and the SNS junction, where occupied (red) and unoccupied (grey) states are aligned in energy by a source–drain bias $V_{SD}$. Without MW irradiation, this gives a differential conductance as the blue curve in Fig. 1b, with peaks corresponding to ABSs at $V_{SD} = \pm (\Delta + E_A)/e$, where $\Delta$ is the superconducting gap and e is the elementary charge. Under global MW irradiation with frequency $f$, photons might couple to both the hybrid JJ and the tunnelling probe used to measure the DOS. Coherent coupling of the electromagnetic field to ABSs in the hybrid JJ might change the eigenstates of the junction, forming Floquet–Andreev states (FASs) at energies $E_A \pm nhf$ ($h$ is the Planck constant and $n$ is an integer), as schematically shown in the middle panel of Fig. 1a. Additionally, PAT in the probe can promote transport across the tunnel barrier by absorption or emission of photons (green) with energy $hf$. The bottom panel of Fig. 1a depicts an example of quasiparticle tunnelling into an ABS assisted by absorption of a photon. Both FASs and PAT give, at least qualitatively, conductance curves as shown in the green trace of Fig. 1b, with peaks at $V_{SD} = (\Delta + E_A \pm nhf)/e$.

A recent work investigated an Al/graphene SNS junction with a tunnelling probe[44]. Conductance replicas of ABSs were observed under MW irradiation and interpreted as signatures of steady FASs in the hybrid JJ. The replicas followed a squared Bessel function power dependence, and summed to a constant value, independent of power— a relation referred to as the sum rule. These features were considered to be additional evidence of FASs. However, coupling between the MW field and the tunnelling probe, which would give rise to PAT, was not considered in ref. 44. It is also well known that PAT gives conductance

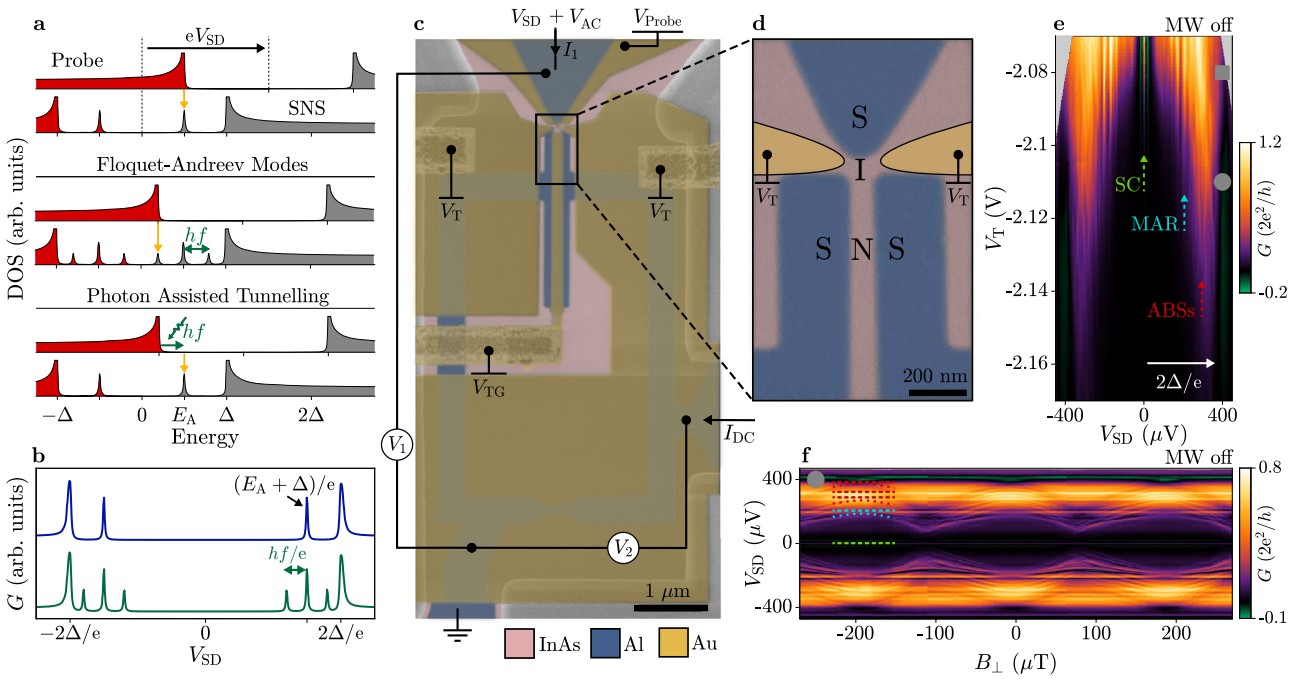

**Fig. 1 | Device under study and tunnelling spectroscopy of sub-gap states.**
**a** Schematic representation of density of states (DOS) and tunnelling spectroscopy into a superconducting–semiconducting–superconducting (SNS) junction using a superconducting weak link (top). Andreev bound states (ABSs) are present in the DOS of the SNS junction at energies $\pm E_A$. A current (yellow) flows when the source–drain voltage $V_{SD}$ aligns occupied (red) to unoccupied (grey) states. In a Floquet–Andreev scenario (middle), replicas of Andreev peaks shifted by the photon energy $hf$ emerge in the DOS of the SNS junctions, giving rise to additional tunnelling resonances. In a photon assisted tunnelling scenario (bottom), absorption of a photon (green) induces tunnelling into an ABSs for $V_{SD} = E_A - hf$. Floquet–Andreev modes are represented as replicas of Andreev peaks shifted by $hf$. **b** Schematic representation of differential tunnelling conductance $G$ measured in the absence (blue) and presence (green) of microwave irradiation. **c** False-coloured electron micrograph of a device identical to that under study, composed of InAs (pink) and Al (blue) and controlled via electrostatic gates (yellow). Gate voltages $V_\alpha$ ($\alpha \in \{T, TG, Probe\}$), bias voltages ($V_{SD}$, $V_{AC}$) and currents ($I_{DC}$), and measured voltages ($V_1$, $V_2$) and currents ($I_1$) are indicated. **d** Zoom-in of the tunnelling junction before gate deposition. The gates controlling the tunnelling barrier transparency are drawn in yellow. **e** Differential conductance $G$ of the tunnelling probe as a function of bias $V_{SD}$ and gate voltage $V_T$. Signatures of the supercurrent (SC, light green), multiple Andreev reflections (MAR, light blue) and ABSs (red) are indicated with dashed arrows. The transport gap is indicated as $2\Delta/e$ (white arrow). **f** Tunnelling spectroscopy of sub-gap states at $V_T = -2.11$V, as a function of perpendicular magnetic field $B_\perp$. Conductance features labelled with colours defined in **e**.

replicas that scale as a squared Bessel function[12, 45], and therefore fulfil the sum rule for any number of replicas.

Distinguishing the generation of FASs from PAT is crucial for engineering light–matter coupling in nanoscale hybrid devices. Despite their similar conductance response, FASs and PAT have drastically different origins. First, FASs are the result of coupling of the MW field to the ABSs in the hybrid JJ, whereas PAT arises within the tunnelling probe, without the requirement of MW irradiation in the JJ. It is also expected that the coupling strength of FASs depends on the ABS properties, in particular the Fermi velocity[44]. This is not the case for PAT, where absorption or emission of photons at the tunnel barrier does not depend on the wave vector of ABSs in the hybrid JJ. Finally, FASs correspond to coherent coupling of ABSs to the MW field, and are therefore expected to induce avoided crossings and to open band gaps in the sub-gap spectrum[1,9,46]. In contrast, quasiparticle tunnelling by photon absorption or emission in the probe does not change the eigenstates of the hybrid JJ, and thus no avoided crossings are expected for PAT.

In this work, we investigate the effect of MW irradiation in the tunnelling spectrum and supercurrent of hybrid JJs in InAs/Al heterostructures. We perform multiple experimental tests to distinguish FASs from PAT and establish that, in our devices, PAT is the dominant phenomenon arising in response to MW irradiation. In the light of our results, conductance replicas which obey a sum rule are compatible with both FASs and PAT, and the additional tests we present here provide tools for their discrimination. Our techniques are applicable to existing devices and will guide towards the establishment of FASs in hybrid nanostructures.

## Results

### Planar superconducting quantum interference device

Figure 1c shows a false-coloured micrograph of the device presented in this Article. The device consisted of a planar superconducting quantum interference device (SQUID) fabricated in a heterostructure of InAs (pink) and epitaxial Al (blue)[42,47,48], covered by a thin HfO₂ insulating layer and with patterned Au gate electrodes (yellow). The superconducting loop contained a planar Al/InAs/Al junction and an Al constriction, all defined in the epitaxial Al. This Al constriction was designed to limit the switching current of the metal arm, while being significantly larger than that of the SNS junction. This configuration allowed for a stable phase drop across the SNS of $\varphi = 2\pi(\Phi/\Phi_0)$, where $\Phi$ is the flux threading the SQUID and $\Phi_0 = h/2e$ is the superconducting flux quantum. The Al loop was connected to two low-impedance superconducting leads, which allowed switching current measurements. A gate-tunable superconducting tunnelling probe was integrated close to the SNS junction, allowing for spectroscopy into the normal region. Two gates controlled the transparency of the tunnelling probe by the gate voltage $V_T$. The SNS junction was controlled by a top gate, which was set to $V_{TG} = -0.8$ V for the results shown in the Main Text, unless otherwise stated. An additional gate was kept to $V_{Probe} = 0$ for the whole experiment. A zoom-in close to the tunnelling probe, obtained prior to gate deposition, is shown in Fig. 1d, where tunnelling gates are shown schematically. Microwave signals were applied via an attenuated coaxial line terminated in an antenna and placed approximately 1cm away from the chip surface. Tunnelling conductance measurements were performed by lock-in techniques. A schematic of the measurement configuration is depicted in Fig. 1c. Tunnelling spectroscopy required sourcing a voltage bias $V_{SD} + V_{AC}$ on the tunnelling probe lead and measuring the resulting AC current $I_1$ and AC voltage $V_1$. The current–phase relation (CPR) was measured by applying a DC current $I_{DC}$ through the loop and measuring the resulting DC voltage $V_2$, which gave the loop switching current. The CPR of the SNS junction was obtained by subtracting the known switching current of the Al constriction from that of the loop. Measurements were performed in a dilution refrigerator at a mixing

chamber base temperature of 7 mK. Further details on materials, fabrication and measurement techniques are reported in the Methods section.

### Tunnelling spectroscopy without microwave irradiation

Figure 1e shows the differential conductance $G \equiv I_1/V_1$ of the tunnelling probe as a function of $V_T$ and $V_{SD}$, as the gate-tunable probe transitioned from the open to the tunnelling regime [top and bottom part of Fig. 1e, respectively]. The open regime was characterised by a zero-bias conductance peak, which represents a supercurrent flowing through the tunnelling probe independent of subgap spectrum in the SNS junction [green dashed arrow, Fig. 1e]. The presence of several finite bias features, emerging below the gap at regular intervals in the open regime, indicate multiple Andreev reflections (MARs) [light blue dashed arrow in Fig. 1e]. The tunnelling regime displayed pronounced features at a voltage $2\Delta/e = 380$ μV (white arrow), consistent with the superconducting gap $\Delta = 190$ μeV of Al[49]. Figure 1f shows $G$ at low barrier transparency ($V_T = -2.11$ V), as a function of perpendicular magnetic field $B_\perp$ and voltage bias $V_{SD}$. Several finite bias conductance peaks are evident in Fig. 1f. Features at high bias with a strong dependence on $B_\perp$ indicate highly transmissive ABSs present within the superconducting gap of the SNS junction, while similar features at low bias are consistent with MAR (light blue dashed lines). In addition, some features in the conductance spectrum (marked by red dashed lines) are attributed to disorder in the tunnelling barrier and sub-gap states in the DOS of the superconducting probe[50].

### Tunnelling spectroscopy with microwave irradiation

The effect of MW irradiation on the tunnelling conductance $G$ for $B_\perp = 0$ is summarised in Figs. 2 and 3. Figure 2 shows $G$ measured with the probe in the tunnelling regime ($V_T = -2.11$ V), in the absence of MW irradiation [Fig. 2a], and as a function of MW source power $P$ at frequencies $f_i = \{4.65, 7.40, 9.20, 12.65\}$ GHz [Fig. 2b–e respectively; see Supplementary Note 1 for a full frequency dependence]. For each frequency, conductance features at finite bias, visible in the top row of Fig. 2, split into replicas as $P$ increased. This is reminiscent of conductance replicas in ref. 44, with a spacing of $hf_i/e$ indicated by the green dashed lines. While performing this analysis, it is important to distinguish conductance peaks that exclusively appear under MW irradiation, to those already present without irradiation and that are caused by sample-specific features such as MAR or sub-gap states in the superconducting probe [see red arrows in Fig. 2a]. In the closed $V_T$ regime, a remnant of the supercurrent is still visible by saturating the colourscale and zooming in close to zero bias, as shown in the bottom row of Fig. 2. Conductance features at zero bias also split into replicas separated by $hf_i/2e$ (green dashed lines), and are readily interpreted as Shapiro steps[12], which occur by photon absorption or emission in the tunnel barrier. Such replicas in the supercurrent flowing through the probe are also clearly visible when the transparency of the tunnel barrier is increased ($V_T = -2.08$ V), as shown in Fig. 3. Crucially, all conductance features split at the same power (see blue arrows) and evolved in an identical fashion as a function of $P$ as $V_{SD} = (hf/e)\alpha$, where $\alpha = \alpha_0 \cdot 10^{P/20}$. This was true for each frequency investigated. An exemplary fit to replicas in the supercurrent at $f = 7.40$ GHz yields $\alpha_0 = 3.0$, which is plotted as the dotted blue lines in Figs. 2c and 3c. Dashed blue lines correspond to an identical coupling strength shifted to high bias, and exactly describe the power dependence of high bias replicas without additional fitting. Independent calculation of $\alpha_0$ from high-bias conductance replicas gives good agreement to values obtained from low-bias replicas (see Supplementary Notes 3 and 7). These results indicate that the coupling strength to high-bias conductance replicas is identical to that of Shapiro steps in the supercurrent across the tunnelling barrier, which is fully consistent with PAT but not expected for FASs.

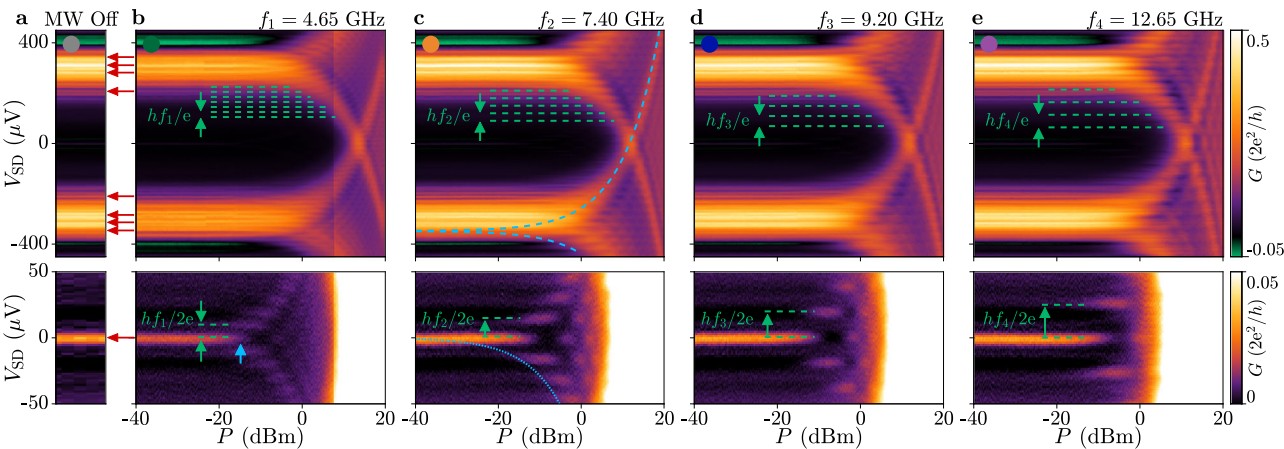

**Fig. 2 | Tunnelling conductance under microwave irradiation in the closed regime. a** Conductance at $V_T = -2.11$ V with no microwave (MW) signal applied. Red arrows indicate sample-specific features present without irradiation. **b**–**e** Conductance at $V_T = -2.11$ V for several irradiation frequencies as a function of MW source power $P$ and $V_{SD}$. The onset of splitting in conductance features is indicated by blue arrows. Blue dotted lines indicate the fitted power dependence of low-bias conductance replicas. Blue dashed lines indicate an identical coupling strength shifted to high bias, which exactly matches high-bias conductance replicas. Zoom-ins close to zero bias highlight remnant supercurrent (bottom). Periodic replication of conductance features is indicated by green dashed lines.

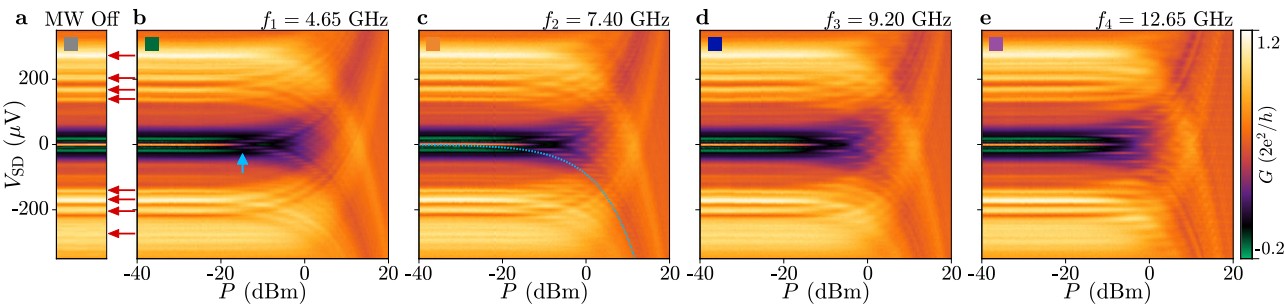

**Fig. 3 | Device conductance under microwave irradiation in the open regime. a** Conductance at $V_T = -2.08$ V with no microwave (MW) signal applied. Red arrows indicate sample-specific features present without irradiation. **b**–**e** Conductance at $V_T = -2.08$ V for several irradiation frequencies as a function of MW source power $P$ and bias $V_{SD}$. The onset of splitting in conductance features (blue arrows) and the power dependence of both low and high bias conductance replicas (dotted and dashed blue lines, respectively) are identical to the equivalent values in Fig. 2.

Selected linecuts of Fig. 2b–e are presented in Fig. 4a, after subtraction of a slowly varying background, together with a periodic grid with spacing $hf_i/e$ (see Supplementary Note 5). Figure 4b summarises the spacing between conductance replicas as a function of MW frequency, obtained from conductance replicas across the full range of power in Figs. 2 and 3, and also includes additional frequencies, another $V_{TG}$ value and a second device (see Supplementary Notes 8 and 9). Conductance replicas at finite bias are indicated by circles and depend on frequency as $\Delta V_{SD} = hf/e$ (dashed line). Supercurrent replicas are indicated as squares and follow the dependence $\Delta V_{SD} = hf/2e$ (dotted line).

An example of conductance replicas at $V_{TG} = -1.4$ V is shown in Fig. 4c. Blue lines indicate a coupling strength $\alpha_0 = 3.0$, identical to that in Figs. 2c and 3c. Decreasing the top-gate voltage from $V_{TG} = -0.8$ V to $V_{TG} = -1.4$ V is expected to reduce the Fermi velocity by ≈25%. In a model of FASs, $\alpha$ is proportional to the Fermi velocity[44], and therefore, assuming that all other parameters in the system stay the same, $\alpha$ is predicted to decrease by the same factor in Fig. 4c relative to Figs. 2c and 3c (yellow lines). However, there is no observed change in the MW coupling strength as a function of $V_{TG}$, consistent with conductance replicas induced by PAT in the tunnel barrier and incompatible with FASs generated in the SNS junction (see Supplementary Note 8 for further details).

Finally, we present the power dependence of conductance replicas shown in Fig. 2. Linecuts of Fig. 2d are shown in Fig. 4d, for $n = 7$ replicas (coloured circles). Their power dependence is modelled by a theory for PAT (lines)[10,45], in which the conductance scales as a squared Bessel function $G \propto J_n^2(\alpha)$. This is similar to the theory used in ref. 44, which follows the same dependence and differs mainly in the definition of $\alpha$. The PAT model takes two input parameters: the low-power conductance [Fig. 2f] and $\alpha_0$, which is calculated from the low-bias conductance replicas (see Supplementary Notes 3 and 4). In the case of $f = 9.20$ GHz, $\alpha_0 = 2.5$. With no free parameters, a good agreement with the data is obtained up to $P \approx 10$ dBm, corresponding to $\alpha = 2.5 \cdot 10^{10~\text{dBm/20}} \approx 8$. This further demonstrates that high-bias conductance features are explained by PAT through the tunnelling barrier, and are not a manifestation of replicas in the DOS of the SNS junction.

We note that the sum over conductance features $S$ is constant for the range of powers where conductance replicas remain within the measured range of bias $V_{SD}$ [see inset of Fig. 4d], similar to ref. 44. This follows the conservation of the tunnelling current, and stems from the result that squared Bessel functions sum to unity. Hence, the sum rule argument cannot be used to distinguish between PAT and FAS interpretations (see Supplementary Note 6 for more details).

Similar to ref. 44, we observed phase modulation of replicas originating from ABSs with energy $E_A = \Delta\sqrt{1 - \bar{\tau}\sin^2(\varphi/2)}$, where $\bar{\tau}$ is the effective junction transmission[19]. Figure 5a–c show the tunnelling conductance as a function of perpendicular magnetic field $B_\perp$ measured under MW irradiation of frequency $f = 9.20$ GHz and applied power $P = -5$, 0 and 5dBm, respectively. Replicas of phase-dependent

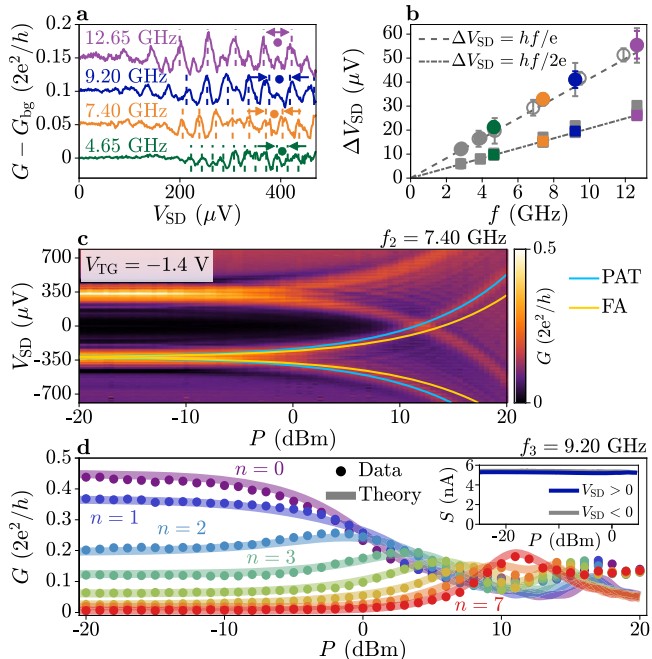

**Fig. 4 | Frequency and power dependence of conductance replicas. a** Linecuts of conductance from Fig. 2, after subtraction of a slowly varying background. Traces are successively offset by $0.05G_0$. Dashed lines mark the expected peak positions. Linecuts are taken at $P = 1.5, 4.5, 4,$ and $4.5$ dBm, respectively. **b** Spacing of conductance replicas measured at finite (circles) and close to zero (squares) bias. Dashed and dotted lines represent the equations $\Delta V_{SD} = hf/e$ and $\Delta V_{SD} = hf/2e$, respectively. Filled and empty grey markers refer to additional data collected on the same device and on a second device, respectively (see Supplementary Notes 8 and 9). **c** Conductance at a top-gate voltage $V_{TG} = -1.4$ V, as a function of microwave (MW) source power for irradiation frequency $f_2 = 7.20$ GHz. Blue lines indicate a MW coupling strength of $\alpha_0 = 3.0$, identical to Fig. 2c, h. Yellow lines indicate a coupling strength $\alpha_0 = 2.25$, reduced by 25% with respect to Fig. 2c, h. Blue (yellow) lines show the expectation for photon assisted tunnelling (Floquet–Andreev states). **d** Conductance of the first seven replicas in Fig. 2i, taken at constant bias $V_{SD}$ (circles), alongside the simulated conductance from a photon assisted tunnelling model (lines, see Supplementary Figs. 3–5). Inset shows the sum $S$ of conductance features in Fig. 2i over positive (blue) and negative (grey) bias. Data is shown for the range of powers where all conductance replicas are within the measured bias range.

ABS features are indicated by green dashed lines, which describe ABSs with transmission $\bar{\tau} = 0.84$. When $P$ was increased, more replicas appeared in the spectrum [Fig. 5b] until replicas originating from positive and negative bias overlapped [Fig. 5c; see Supplementary Note 10 for further details]. No avoided crossing was observed for overlapping conductance features, in disagreement with FAS predictions[1, 9, 37].

In summary, conductance replicas appearing under MW irradiation are fully compatible with PAT through the tunnelling junction used to probe the DOS, and incompatible with FASs. This was concluded from the power dependence of the conductance, which was identical to that of the Shapiro steps, the absence of a dependence on the Fermi velocity in the planar JJ, and the absence of avoided crossings at zero energy.

## Current–phase relation under microwave irradiation

After demonstrating that spectral replicas at high bias are caused by PAT in the tunnelling junction used to perform spectroscopy, we investigate how ABSs in the SNS junction couple to the applied electromagnetic field. These experiments probe the macroscopic superconducting state and do not rely on the microscopic processes taking place within the tunnelling probe, which was left floating. In particular, each occupied ABS in the SNS carries a supercurrent

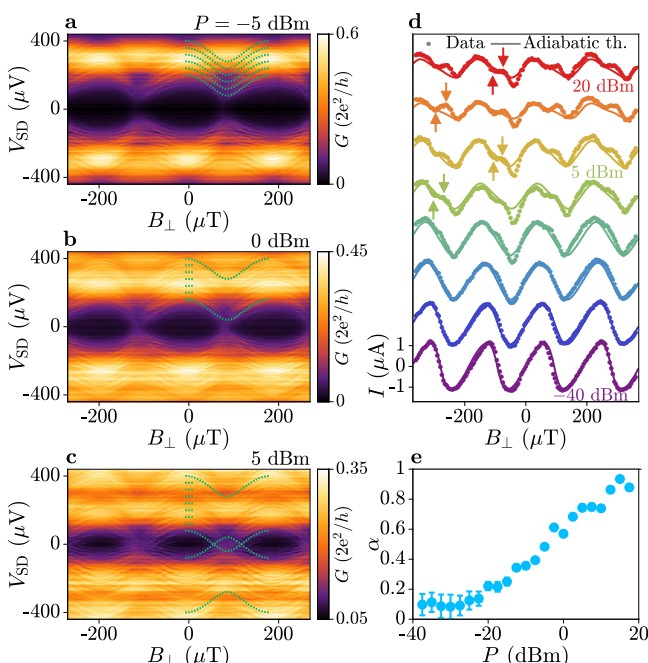

**Fig. 5 | Phase-dependent measurements under microwave irradiation.**
**a** Tunnelling conductance as a function of perpendicular magnetic field $B_\perp$ for microwave (MW) irradiation with frequency $f = 9.20$ GHz and power $P = -5$ dBm. Green dashed lines mark bound state replicas with effective transmission $\bar{\tau} = 0.84$. **b, c** Same as **a** for $P = 0$ dBm and $P = 5$ dBm, respectively. **d** Current–phase relation as a function of MW power (circles) fitted with an adiabatic theory (lines), for MW frequency $f = 9.20$ GHz. Traces are successively offset by $2.05$ μA. Deviations of the data from the adiabatic theory at high power are marked with arrows. **e** Microwave field strength $\alpha$ obtained from the adiabatic theory fit presented in **d**, with error bars showing the uncertainty in the fit parameter to one standard deviation. Data was taken at $V_{TG} = -0.8$ V and $V_T = -2.11$ V.

$I = -(2e/\hbar)[\partial E_A(\varphi)/\partial\varphi]$. The total supercurrent flowing in the SNS is obtained by summing the contributions of each ABS[42]. Figure 5d shows the CPR of the SNS junction as a function of MW power. For $P < -20$ dBm, we observed a forward-skewed CPR, which indicates the presence of highly-transmissive ABSs[19,42, 51], consistent with the spectrum in Fig. 1f.

Increasing the applied MW power, both the amplitude and skewness of the CPR decreased. This behaviour is described by an adiabatic theory of ABSs with a time-varying phase $\phi(t) = \varphi + 2\alpha\cos(2\pi ft)$, where the electromagnetic field strength is $\alpha = eV_{MW}/hf$[52, 53]. In this framework, the adiabatic current is given by $I_{ad.} = \Sigma_n J_0(2n\alpha)I_n(\varphi)$, where $J_0$ is a Bessel function of the first kind and $I_n(\varphi)$ are the experimentally-determined harmonics of the CPR at equilibrium[52]. A fit with $\alpha$ as the sole free parameter describes the data well (solid lines), with fitted values of $\alpha$ shown in Fig. 5e (see Supplementary Notes 11 and 12 for more details). At $P = 0$ dBm we extract $\alpha_0 = 0.6$, significantly smaller than $\alpha_0 = 2.5$ obtained for $f_3 = 9.20$ GHz from low-bias replicas in Fig. 2d (see Supplementary Note 3). This indicates that the coupling strength of the MW field to the ABSs in the SNS junction, which is the parameter controlling the formation of FASs, is much smaller than that extracted from spectral replicas. This discrepancy is fully consistent with a PAT origin of spectral replicas, not linked to processes taking place in the SNS junction.

The adiabatic theory describes the data well up to an applied power of $P \approx 5$ dBm, corresponding to $\alpha \approx 0.6$. For larger $P$, the adiabatic model still captures the CPR envelope, but does not account for dips in the CPR appearing at specific values of $B_\perp$ [arrows in Fig. 5d]. Supercurrent dips are explained by non-equilibrium ABS occupation due to absorption or emission of MW photons[52, 54, 55]. Transitions occur

when the photon energy $hf$, or integer multiples of it, matches the separation between two ABSs or between an ABS and the continuum. Once a transition occurs across the gap, the newly occupied ABS contributes to the total supercurrent with opposite sign with respect to the ground state, resulting in a dip in the CPR. Dips are resolved in the CPR for $\alpha > 0.6$, consistent with a model of non-equilibrium ABS distribution in a JJ containing highly-transmissive modes (see Supplementary Note 13). Therefore, the dominant effects on the CPR in our devices are an adiabatic modulation of the phase and a non-equilibrium distribution function of ABS occupation, up to $\alpha \gtrsim 1$. It is possible that supercurrent signatures of FASs would emerge for larger MW powers, but conductance features would be masked by the much stronger PAT effects.

## Discussion

In summary, conductance replicas were realised in a hybrid JJ with highly transmissive ABSs under MW irradiation. They obeyed a sum rule, consistent with both FASs in the junction and PAT in the probe. By careful investigation of the replicas, the electromagnetic field was shown to dominantly affect transport through the superconducting probe. Consequently, we concluded that conductance replicas primarily originated from PAT, an effect not considered in ref. 44. Specifically, the experimental tests we performed gave results inconsistent with a theory for FASs. First, the power dependence of conductance replicas was identical to that of Shapiro steps in the tunnelling junction, whereas a difference is expected for FASs. Second, the coupling strength $\alpha$ associated with conductance replicas was significantly larger than that associated with ABSs in the SNS junction and measured via switching currents, but should be equal in the case of FASs. Third, the coupling strength was independent of the Fermi velocity, inconsistent with the linear dependence predicted for FASs. Fourth, conductance replicas brought to zero energy crossed each other, while avoided crossings are expected for FASs. Complementary measurements of the CPR of the JJ are consistent with an interaction between ABSs and the MW field mediated by the superconducting phase difference, without the need to invoke FASs. The weak coupling of the MW field to ABSs is presumably due to the use of an off-chip MW antenna, which predominantly interacts with the device via the large leads. Future work can engineer more efficient coupling schemes, for example by applying local MW signals via gate electrodes[24], enabling stronger interaction with ABSs while limiting heating in the setup.

Our results show that caution should be used to attribute replicas in the tunnelling conductance to the presence of FASs in hybrid JJs. However, the techniques outlined here constitute a baseline to evaluate the effect of light–matter interaction in nanoscale devices, as they give distinct signatures for FASs and PAT, and can be applied in generic cases.

## Methods

### Materials and fabrication

The devices under study were fabricated from a heterostructure grown on an InP (001) substrate by molecular beam epitaxy techniques. The heterostructure consisted of a step-graded metamorphic InAlAs buffer and a 8 nm thick InAs quantum well, confined by $In_{0.75}Ga_{0.25}As$ barriers 13 nm below the surface. A 15 nm thick Al layer was deposited on top of the heterostructure, in the same chamber as the III–V growth without breaking vacuum. The peak mobility in a gated Hall bar was 18000 $cm^2V^{-1}s^{-1}$ for an electron density of $n = 8 \cdot 10^{11} cm^{-2}$. This gave an electron mean free path of $l_e \gtrsim 270$ nm, hence we expect all Josephson junctions measured here to be ballistic along the length $L$ of the junction.

Devices were fabricated by first isolating large mesa structures in the III–V material to prevent parallel conduction between devices. This was done by selectively removing the top Al layer with Al

etchant Transene D, before etching $\approx 350$ nm into the III–V heterostructure using a chemical wet etch (220 : 55 : 3 : 3 solution of $H_2O : C_6H_8O_7 : H_3PO_4 : H_2O_2$). The planar SQUID device was then patterned on top of the mesa structure, by selective etching of the Al with Transene D at 50°C for 4 s. We deposited a dielectric by atomic layer deposition, consisting of a 3 nm $Al_2O_3$ layer below 15 nm of $HfO_2$, before evaporating metallic gate electrodes to control the exposed III–V regions. These were deposited in two steps: fine gate features above the planar SQUID were first defined with 5 nm of Ti and 20 nm of Au; these were contacted with 10 nm of Ti and 400 nm of Al, to connect the mesa structure to the bonding pads.

### Measurement techniques

Measurements were performed in a dilution refrigerator with a base temperature of 7 mK. Conductance measurements were performed with standard lock-in-amplifier techniques. An AC voltage $V_{AC} = 3 \mu V$ was applied to a contact at the superconducting probe with a frequency of 311 Hz. The current flowing through the probe to ground, $I_1$, and the differential voltage across the tunnel barrier $V_1$ were measured to give the conductance $G \equiv I_1/V_1$. The transmission from the probe to the SNS junction was tuned using the gates $V_T$. In the tunnelling regime, where the conductance is lower than one conductance quantum $G_0 = 2e^2/h$, the measured conductance is proportional to the convolution of the density of states in the probe and in the junction. A constant bias offset of 43 μV was subtracted from all datasets, due to a DC offset at the current–voltage (I–V) converter. The plotted bias voltage $V_{SD}$ at the device was adjusted from the sourced value to account for the voltage dropping across a measured line resistance of 5.8 kΩ. Current-biased measurements were performed on the same device. Both contacts at the superconducting probe were floated, such that no current flowed through the probe. A DC current $I_{DC}$ was applied symmetrically to the SQUID loop, such that the potential of the device was not raised with respect to the gate electrodes. A sawtooth current signal was applied from a waveform generator at a frequency of 133 Hz. The voltage drop across the SQUID loop $V_2$ was measured with an oscilloscope. Once the voltage passed a threshold signifying a superconducting–resistive transition, the switching current was recorded. To account for stochastic phase escape behaviour, the average switching current was recorded over 36 values. A high-frequency signal was applied to the device via an antenna ~1 cm away from the surface of the chip. The antenna, an exposed coaxial line, was attached to a MW line with attenuation 47 dB. Powers $P$ refer to the power outputted at the signal generator.

## Data availability

The raw and processed data generated in this study are available in the Zenodo database [https://doi.org/10.5281/zenodo.8298619]. Further data that support the findings of this study are available upon request from the corresponding author.

## Code availability

The computer code used to obtain the results presented in this study is available in the Zenodo database [https://doi.org/10.5281/zenodo.8298619].

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

## Acknowledgements

We are grateful to C. Bruder, W. Riess and H. Riel for helpful discussions. We thank the Cleanroom Operations Team of the Binnig and Rohrer Nanotechnology Center (BRNC) for their help and support. F.N. acknowledges support from the European Research Council (grant number 804273) and the Swiss National Science Foundation (grant number 200021_201082). W.B. acknowledges support from the European Union's Horizon 2020 FET Open programme (grant number 964398) and from the Deutsche Forschungsgemeinschaft (DFG; German Research Foundation) via the SFB 1432 (ID 425217212). J.C.C. thanks the Spanish Ministry of Science and Innovation (Grant No. PID2020-114880GB-I00) for financial support and the DFG and SFB 1432 for sponsoring his stay at the University of Konstanz as a Mercator Fellow.

## Author contributions

F.N. conceived the experiment. E.C., F.K., R.S. and W.W. performed the material synthesis and characterisation. D.Z.H. and F.N. designed the samples. D.Z.H. and M.C. fabricated the samples. D.Z.H. performed the experiments, M.C. and F.N. provided support. D.Z.H. analysed the data, W.B. and J.C.C. provided theoretical support. J.C.C. analysed the non-thermal distribution in the current–phase relation. D.Z.H., D.S., W.B., J.C.C. and F.N. interpreted the data, with contributions from M.C., M.H. and S.C.t.K. D.Z.H. wrote the manuscript with input from D.S. and F.N. and with contributions from all authors.

## Competing interests

The authors declare no competing interests.
