## [Peer Review File · Nature Communications]

Reviewers' Comments:

Reviewer #1:

Remarks to the Author:

In this manuscript, Haxell et al. report on a detailed analysis of experimentally recorded microwave-induced conductance replicas in a nano-structured Josephson junction (JJ). The main point of the work is the experimental distinction between Floquet-Andreev states (FAS) and photon-assisted tunneling (PAT). The former was claimed to be observed by Park et al. (Ref.10) in a graphene-based device, and in this paper, a thorough analysis leads the authors to conclude that it is instead PAT, which is causing the observed conductance replicas arising in a 2D InAs/Al hybrid device, and that the distinction between the two mechanisms requires a deeper analysis of the data.

The experiment records nonlinear conductance vs. bias voltage and a tunnelling-probe gate voltage, V_T , which tunes the transmission of the JJ from open to more closed. By irradiating the sample by microwave radiation, evenly spaced sidebands are observed of the narrow supercurrent peak as well as off the finite-bias conductance peaks present without the radiation, with the expected spacings of respectively $f/2e$ and f/e .

Changing the gate voltage, V_{TG} , adjusts the electron density in the normal region of the SNS JJ. Changing from $V_{TG} = -0.8V$ to $V_{TG} = -1.4V$ is estimated to change the Fermi velocity by some 25%, which in turn would change the coupling strength to the microwaves in a FAS scenario. As seen in Fig. 3c, however, the power law dependence remains on top of the blue line, and the coupling strength is therefore the same for the two different top gate settings, i.e. densities (which are estimated by switching current and separate Hall bar measurements), rendering a FAS scenario less likely. Also, both the (Shapiro) sidebands of the supercurrent peak and those of the finite bias peaks follow the same dependence on power, which should not be the case for FAS. Fitting the CPR in Fig. 4d by an adiabatic approximation gives rise to a much smaller coupling, which is interpreted as the microwaves coupling much weaker to the ABS in the SNS junction than to the superconducting leads as in PAT. Also, the lack of avoided crossings in Fig. 4c is stated to disagree with FAS.

Overall, this is a timely piece of work, with very high-quality data and a very thorough analysis. The text is well written, although, as indicated below, the reader could be assisted more here and there. This is a different device from Ref. 10, but the message told here is of a general nature, namely how to interpret the data and distinguish between PAT and FAS. The distinction made here is convincing. It rests on a few uncertainties, but altogether, I am left convinced that this is a nice illustration of PAT.

Nevertheless, the physical question at hand seems to be how well the photons are absorbed respectively by the quasiparticles in the leads or the ABS on the junction, or plainly to what extent the microwaves drive the leads or the top gate (or possibly the normal region by itself). As stated in Venitucci et al. [PRB 97, 195423 (2018)], a time-dependent potential in the leads can to some extent be gauge transformed to a time-dependent gate-voltage. Therefore, the distinction between PAT and FAS is not entirely clear. This work nicely shows how to analyze the data, and it shows that both quasiparticles, Cooper pairs and ABS feel the microwaves, but that photon assisting the two former is what dominates the bias spectroscopy. It might however be more instructive (and of greater general impact) to aim the story towards a more general discussion about how the microwaves are absorbed in these devices, or rather, what experiments can tell us about this. This would not require a major rewriting, but rather an adjustment of the manuscript towards a fusion of PAT and FAS into a more general scenario. Taking its outset in the acclaimed FAS in Ref. 10, as is done already, and questioning this physical picture invoked by the convincing PAT analysis. PAT has a longer history, and 'Floquet (engineering)' might have a touch of novelty to it, but the sidebands, the Bessel functions and the photon assisting remains the same (which is also the spirit of Ref. [20] (Peters et al.) and of Venitucci et al. mentioned above), and the interesting physics question, which will aid real progress with these devices, appears to be how and where the photons are absorbed, and which electrons are being assisted the most by which design. For this, a thorough analysis like the one carried out here is necessary, and that is definitely worth pointing out.

If the authors agree with this angle to the story, I could recommend publication of a modified version of the manuscript, which is directed away from the PAT/FAS dichotomy and made more general in this way, if the questions and comments stated below can be satisfactorily answered. If the authors disagree with this more general perspective, I am of course interested in hearing why,

but then the manuscript appears more as a correction to Ref. 10, pointing out that the FAS story told there is not the full story of such devices, which I find to have less constructive impact in the community.

Comments/Questions:

- 1) How should one understand the finite-bias conductance peaks in Figs. 2a and 2f (and Fig. 1e)? The paragraph on top of 2nd column of page 2, leaves the impression that they are a mix of multiple Andreev reflections (MAR) and Andreev bound states (ABS) induced by the probing superconductor. The V_T dependence (and their spacing) in Fig.1e suggests that it is not MAR (it seems?), but what causes the very regular gate dependence observed in Fig.1e, and is the induced gap itself V_T dependent?
- 2) How does Fig.1e look for $V_{TG} = -1.4V$, where Fig.3c is recorded? Is this available?
- 3) In Fig.3b, where are the squared symbols measured? One identifies very easily the points in Fig.2 with the gate settings in Fig.1e, but this seems not to be possible with the points in Fig. 3b, where one can capture the circles from Fig. 3a, but not the squares, corresponding to supercurrent sideband spacings?
- 4) How sensitive is the background subtraction in Fig. 3a to the choice of averaging bias window of $70 \mu V$, given in the supplementary?
- 5) The fits in Fig.2c gives rise to $\alpha_0 = 3.0$. The caption only refers to the dotted line, and it would be useful to also refer explicitly to the dashed. As I understand it, both are fitted by the same function for the splitting with $\alpha_0 = 3.0$, is that correct? Same for the fits (dotted and dashed) in Fig.2h, right?
- 6) The fit in Fig.3d is said on p.3 to give $\alpha \approx 8$ at $P \approx 10$ dBm. Should this be understood by taking the $\alpha_0 = 3.0$ and the $V_0 = 91 \mu V$ for 7.4 GHz, using the formula in the supplementary for $\alpha_0 = eV_0/hf$ to convert to $\alpha_0 = (3.0 * 7.4 * 96 / (9.2 * 91)) 10^{(10/20)} \approx 8$? If yes, then perhaps help the reader a bit more in the main text, if no, then what did I misunderstand? The point seems to be, that parameter V_0 , encoding the coupling, the line damping and the output antenna voltage, depends on frequency. Is that correct? But how does this demonstrate (as stated) that the high bias conductance features are explained by PAT with the same coupling strength as to the low bias features? Is that because the fit in Fig.3b gives $\alpha_0 = 3.0$, as found also for Fig.2c? This is a very important point, and I think the reader is better convinced with a little more assistance.
- 7) At which value of V_T is Fig.4 recorded (please state in the Figure caption)? My guess is that it is at $V_T = -2.11 V$, since this is where Fig.3S refers to and where I know that $\alpha_0 = 2.5$ (why 2.6 ± 0.1 as stated in p.5, col.1, line 4?).
- 8) In Fig.4e, how does this scatter plot compare to the power law formula used before? Perhaps show them together?
- 9) In supplement section 8, first line of third paragraph "...remarkable agreement with those at $0.8V$ ", I suppose there should be a minus in front of 0.8 ?
- 10) In supplement formula (S.3), should the " $=0$ " be removed in the first argument of G on the right-hand side?

Reviewer #2:

Remarks to the Author:

This manuscript presents an in-depth study of the behaviour of semiconductor/superconductor Josephson junctions under microwave radiation. In particular, the authors describe the delineation between two effects of different origin which result in qualitatively similar behaviour in tunnelling spectroscopy (Figs 1a/b). The first effect is photon-assisted-tunneling (PAT), whereby photon absorption enables otherwise prohibited transport at energies in integer multiples of hf/e above/below pre-existing features in the density of states, while the second effect is the possible generation of Floquet-Andreev states (FAS), a radiation-induced modification of the density of states itself. While fundamentally different processes, the basic transport signatures are strikingly similar, down to the Bessel-like power-dependence of the conductance of the hf/e 'replicas'. A previous publication, ref10, claims the observation of FAS without ever considering PAT, despite PAT being a generic phenomenon in all two-terminal electronic devices, and routinely presented in textbooks on superconductivity and quantum transport. The current manuscript is therefore extremely timely, since it provides a much more detailed look at possible behaviours of Josephson junctions under radiation than reference 10, and provides a methodology for distinguishing

between PAT of ABSs and FAS.

I can find no major error in analysis and find the experimental work of the highest standard. In particular, the data are consistently of superior quality compared to that of reference 10, likely due to their ability to keep their device cold while under radiation; sample heating seems to have been a big problem in reference 10.

However, I have several suggestions to the authors regarding the presentation of the data. I found the manuscript a little confusing and some of the data is challenging for a new reader to interpret in the current presentation. There seems to be two major issues: 1) the proposed scheme to discriminate between FAS and PAT is not clear to the reader until the discussion, even though this scheme could have been presented 'as known' from previous literature. 2) The first time the reader encounters power-dependent spectra in Fig 2, the data are very, very complicated, with signatures from the supercurrent, multiple ABSs, and multiple Andreev reflections (MARs). As the authors note, all these features produce replicas under applied microwave radiation, meaning that the data they present in Figure 2 is a mix of various different signatures, which is impossible for a first-time reader to decode. Even when the authors add guides, sometimes these get in the way; e.g. the green lines should be on the right side of the sub-figures in my opinion, since they risk obscuring important data. Conversely, I actually think the authors could have added guides in Fig 1e/f to immediately help the reader understand which features are which.

To fix these issues, I would suggest first presenting data with no supercurrent, minimal MARs, and a single (or low number of) ABS, and show how this single ABS evolves under microwaves before presenting any of the figure 2 data. That is, the authors should show in Figure 1 that they can effectively reproduce the result of ref 10; I think most readers will be happy to see this early on. The data from Fig 3c, or many of the sub-figures of supplementary figs S9, S10, S11 could serve. Once this 'baseline' is established, the data in figure 2 will be much easier to understand.

Here is a suggested flow:

- 1) As is for Fig 1 a-e.
- 2) Fig 1f is replaced by data from Fig 3c/S9-11, i.e. 'reproducing reference 10'. The current Fig 1f can be moved to the top of Figure 4, where the reader will want it anyway to compare with the rest of the sub-figures.
- 3) Use a paragraph to clearly outline the proposed delineation scheme.
- 4) Present the rest of the data basically as is, but now the authors will be able to reference everything back to the proposed discrimination scheme, which the reader will now already know about.

If the authors can rework the manuscript to make it a little more accessible to readers, it will be more than suitable for publication in Nature Communications. In fact, I don't see why it shouldn't be published alongside reference 10 in Nature itself.

Minor comments:

- The references are out of order.
- In Fig 1e, the features 'bend' at higher gate voltages. Is this simply due to not correcting the V_{sd} axis for the lowered device resistance? If so, I would suggest correcting for this, so readers understand that the gap, MARs etc should be energy independent.
- Consider increasing the size of Fig 2, or split the data over two figures, so that the features are more clearly visible, especially on a printed page.
- When the authors are discriminating between features, there are basically five phenomena they could be: ABSs, MARs, the gap itself, supercurrent or Yu-Shiba Rusinov states (YSR not so likely given no other evidence of a quantum dot or spin impurity). Each of these presents either at different energy and/or has a different radiation power dependence. Perhaps this is worth stating explicitly, since I presume this is how the authors were able to distinguish between the features.

And finally, out of interest: Have the authors raised the issues with the editors of Nature and/or the authors of reference 10? Since PAT in Josephson junctions has been studied for many decades now (as the second last author here well knows!), it's not feasible for reference 10 to remain published as-is. They really must reanalyse their data to find the extent to which it can be

explained by PAT.

Reviewer #1

[...] Overall, this is a timely piece of work, with very high-quality data and a very thorough analysis. The text is well written, although, as indicated below, the reader could be assisted more here and there. This is a different device from Ref. 10, but the message told here is of a general nature, namely how to interpret the data and distinguish between PAT and FAS. The distinction made here is convincing. It rests on a few uncertainties, but altogether, I am left convinced that this is a nice illustration of PAT.

We thank the Referee for their thorough reading of our work, and their encouraging comments regarding its quality. We agree that our manuscript presents a general approach to distinguish between different phenomena arising from interactions with microwave irradiation.

Nevertheless, the physical question at hand seems to be how well the photons are absorbed respectively by the quasiparticles in the leads or the ABS on the junction, or plainly to what extent the microwaves drive the leads or the top gate (or possibly the normal region by itself). As stated *fx.* in Venitucci et al. [PRB 97, 195423 (2018)], a time-dependent potential in the leads can to some extent be gauge transformed to a time-dependent gate-voltage. Therefore, the distinction between PAT and FAS is not entirely clear. This work nicely shows how to analyze the data, and it shows that both quasiparticles, Cooper pairs and ABS feel the microwaves, but that photon assisting the two former is what dominates the bias spectroscopy. It might however be more instructive (and of greater general impact) to aim the story towards a more general discussion about how the microwaves are absorbed in these devices, or rather, what experiments can tell us about this. This would not require a major rewriting, but rather an adjustment of the manuscript towards a fusion of PAT and FAS into a more general scenario. Taking its outset in the acclaimed FAS in Ref. 10, as is done already, and questioning this physical picture invoked by the convincing PAT analysis. PAT has a longer history, and ‘Floquet (engineering)’ might have a touch of novelty to it, but the sidebands, the Bessel functions and the photon assisting remains the same (which is also the spirit of *fx.* Ref. [20] (Peters et al.) and of Venitucci et al. mentioned above), and the interesting physics question, which will aid real progress with these devices, appears to be how and where the photons are absorbed, and which electrons are being assisted the most by which design. For this, a thorough analysis like the one carried out here is necessary, and that is definitely worth pointing out.

We thank the Referee for their constructive comments on how to increase the impact of our work, and we agree that a more general discussion of microwave absorption in planar Josephson junctions better highlights the significance of our results. As a result, we have made changes to the abstract, introduction, figures, discussion, and conclusions of the manuscript.

We first introduce the different mechanisms by which microwave irradiation couples to superconducting systems, and how they depend on specifics of the device and the way that microwaves are applied. We highlight the problem that two distinct effects, PAT and FASs, which arise from coupling to the probe and the ABSs respectively, can both give rise to sidebands in conductance features with a Bessel function power dependence and constant conductance sum. The work by Park et al. [new Ref. 44] is introduced only after a major introduction. We then note possible distinguishing features of the two phenomena, and how they might manifest in an experiment. We refer to these distinguishing features throughout the manuscript, to clearly state the ways in which we identify PAT as the dominant mechanism for conductance replicas. We then make a more general conclusion, stating that our approach allows us to distinguish how and where photons are absorbed in our device.

If the authors agree with this angle to the story, I could recommend publication of a modified version of the manuscript, which is directed away from the PAT/FAS dichotomy and made more general in this way, if the questions and comments stated below can be satisfactorily answered. If the authors disagree with this more general perspective, I am of course interested in hearing why, but then the manuscript appears more as a correction to Ref. 10, pointing out that the FAS story told there is not the full story of such devices, which I find to have less constructive impact in the community.

We believe that our updated manuscript gives a more impactful discussion of interactions between superconducting systems and electromagnetic irradiation. As such, our results are presented in a general context, rather than as a specific response to [new Ref. 44, Park et al. Nature 603, 421-426 (2022)]. We thank the Referee for this suggestion. We additionally address the comments and questions of the Referee, in the point-by-point response below.

Comments/Questions:

1) How should one understand the finite-bias conductance peaks in Figs. 2a and 2f (and Fig. 1e)? The paragraph on top of 2nd column of page 2, leaves the impression that they are a mix of multiple Andreev reflections (MAR) and Andreev bound states (ABS) induced by the probing superconductor. The V_T dependence (and their spacing) in Fig. 1e suggests that it is not MAR (it seems?), but what causes the very regular gate dependence observed in Fig. 1e, and is the induced gap itself V_T dependent?

We assign conductance features to distinct origins, based on their position in energy and their dependence on tunnel barrier voltage and perpendicular magnetic field. In addition to supercurrent remnants at zero bias, there are multiple conductance features at finite bias. We first note that the original Fig. 1e did not account for the voltage drop across the resistive lines, resulting in an artificial gate dependence of finite-bias conductance features. We have accounted for this error in the updated figure below, where the position of MAR features is now independent on gate voltage. We have updated all tunnelling spectroscopy figures throughout the manuscript. We note that this does not change the conclusions of the paper, nor the results of the analysis.

New Fig. 1 of the Main Text.

In addition, for more negative values of V_T we see a wide band of conductance features. As a function of perpendicular magnetic field B_{\perp} , some features are dependent on B_{\perp} and some are constant. We attribute B_{\perp} -independent features to resonances in the tunnelling probe, while B_{\perp} -dependent features have a characteristic dispersion of ABSs in the planar junction. In Fig. 1f, we also see replicas in B_{\perp} -dependent features without microwave irradiation. Such features are typically produced by coupling to a resonance in the tunnelling probe, as discussed in Ref. 50 [Su et al. Phys. Rev. Lett. 121, 127705 (2018)].

In the new version of the paper, we largely improved the presentation of experimental data to gently introduce all such features to the reader. First, we have added to the discussion in the text, to specifically describe the different conductance features in Figs. 1e and f to clarify the origin of conductance features in the spectrum in absence of microwave irradiation. Second, we label different conductance features in Fig. 1e and f. Third, we split the old Fig. 2 into two parts, now presenting conductance replicas in the tunnelling regime before the more complex spectrum when the tunnel barrier was more open. Fourth, we have modified the labels in Figs. 2 and 3 so that conductance replicas are not obscured. Finally, as in the first version, we distinguish features which are present in the absence of microwave irradiation from additional conductance replicas which emerge for increasing microwave power, using the red arrows in the new Figs. 2 and 3.

2) How does Fig.1e look for $V_{TG} = -1.4V$, where Fig.3c is recorded? Is this available?

We show below the conductance as a function of V_T , at $V_{TG} = -1.4 V$, consistent with Supplementary Figs. 12-15. This figure has been added to the Supplementary Material as a new Supplementary Figure 10, with a short discussion (page 8).

3) In Fig.3b, where are the squared symbols measured? One identifies very easily the points in Fig.2 with the gate settings in Fig.1e, but this seems not to be possible with the points in Fig. 3b, where one can capture the circles from Fig. 3a, but not the squares, corresponding to supercurrent sideband spacings?

The squared symbols are measured using conductance replicas in low bias features. These are taken from the tunnelling regime (bottom panels of the new Fig. 2), and from the more open tunnel barrier regime (new Fig. 3). The separation between conductance replicas was calculated across the full range of powers at which replicas are visible, for both low and high bias replicas. The data in the new Fig. 4a is used to illustrate an example of conductance replicas at high bias.

An example of a linecut of low bias conductance replicas is shown here, to illustrate the frequency dependence. Linecuts are taken from the new Fig. 3 (more open barrier regime, $V_T = -2.08$ V) at a microwave power of $P = -10$ dBm, at the frequencies indicated by the colours (see new Fig. 4b for definition, grey corresponds to the conductance without microwave irradiation).

Fig.: Conductance replicas at low bias, at $V_T = -2.08$ V under microwave irradiation of power $P = -10$ dBm. Colours indicate the microwave frequency; 7.40 GHz (yellow), 9.20 GHz (blue), 12.65 GHz (purple). The grey trace corresponds to the conductance without microwave irradiation.

We now more clearly state how the values in the new Fig. 4b are obtained, both in the Main Text (page 5, left column, second paragraph) and the Supplementary Material (last paragraph on page 6).

4) How sensitive is the background subtraction in Fig. 3a to the choice of averaging bias window of $70 \mu\text{V}$, given in the supplementary?

The averaging bias window was chosen to be slightly larger than the maximum separation of conductance replicas ($\sim 50 \mu\text{V}$ for 12.65 GHz). If the bias window was too small, replica features were considered as part of the background conductance and therefore were excluded from the plot of $G - G_{\text{bg}}$. If the bias window was too large, the background variations were not properly accounted for. Examples of these two situations are shown below.

New Supplementary Fig. 7. Frequency of 4.65 GHz.

New Supplementary Fig. 8. Frequency of 12.65 GHz

We have added a discussion to Supplementary Note 5 to explicitly justify the averaging bias window of $70 \mu\text{V}$ (page 6). We have also included the figures above as Supplementary Figs. 7 and 8. We state that the background subtraction was insensitive to the bias window size for values between hf/e (due to the splitting of conductance replicas) and $250 \mu\text{V}$ (due to the background conductance).

5) The fits in Fig.2c gives rise to $\alpha_0=3.0$. The caption only refers to the dotted line, and it would

be useful to also refer explicitly to the dashed. As I understand it, both are fitted by the same function for the splitting with $\alpha_0=3.0$, is that correct? Same for the fits (dotted and dashed) in Fig.2h, right?

The fits in the new Fig. 2c give rise to $\alpha_0 = 3.0$, which corresponds to the dotted line. The dashed line shows an identical coupling strength shifted to higher bias. The high-bias conductance replicas are well described by the dashed lines without additional fitting, indicating an identical coupling of low and high bias conductance replicas. The same α_0 describes low-bias conductance replicas in the new Fig. 3c (dotted line), up to large microwave powers.

We have updated the caption and the text to state this point clearly. In particular, we emphasise that the dotted lines result from a fit to low bias conductance replicas, and dashed lines are exact copies shifted to high bias and describe the high bias conductance features well without additional fitting (page 5 of Main Text, left column, first paragraph).

6) The fit in Fig.3d is said on p.3 to give $\alpha \approx 8$ at $P \approx 10$ dBm. Should this be understood by taking the $\alpha_0=3.0$ and the $V_0=91 \mu\text{V}$ for 7.4 GHz, using the formula in the supplementary for $\alpha_0=eV_0/hf$ to convert to $\alpha_0=(3.0*7.4*96/(9.2*91))10^{(10/20)} \approx 8$? If yes, then perhaps help the reader a bit more in the main text, if no, then what did I misunderstand? The point seems to be, that parameter V_0 , encoding the coupling, the line damping and the output antenna voltage, depends on frequency. Is that correct? But how does this demonstrate (as stated) that the high bias conductance features are explained by PAT with the same coupling strength as to the low bias features? Is that because the fit in Fig.S3bf gives $\alpha_0=3.0$, as found also for Fig.2ch? This is a very important point, and I think the reader is better convinced with a little more assistance.

We thank the referee for their questions regarding this important point, which we will address in detail.

First, the data (circles) in the new Fig. 4d is stated to agree with the model (lines) up to a power of $P \approx 10$ dBm. This corresponds to a coupling strength $\alpha \approx 8$, using the formula $\alpha = \alpha_0 \cdot 10^{P/20}$ written in the first paragraph of the left column of page 5 in the Main Text. This calculation uses the value $\alpha_0 = 2.5$, since the measurement was performed at 9.20 GHz. We have now added these details to the text, to aid the reader in understanding this calculation (page 5 of the Main Text, right column, first paragraph).

Second, the referee is correct that the parameter V_0 encodes the coupling, line damping and output antenna voltage. This value is directly related to α_0 by $V_0 = (hf/e)\alpha_0$. The specific values of α_0 depend on frequency, presumably due to different resonances in the sample space and the specific shape of the device.

Third, the referee is correct to identify the crucial point that the high and low bias conductance features have the same coupling strength. This can be seen from the following:

1. In the new Figs. 2c and 3c, where a fit to low bias replicas (dotted lines, yielding $\alpha_0 = 3.0$) exactly describes the power dependence of high bias replicas (dashed lines).
2. The new Fig. 4d shows the results of a model for conductance replicas based on the coupling strength α_0 obtained from low bias replicas. The model describes the high bias replicas well without free parameters, showing that low and high bias replicas have the same coupling strength.

3. An independent calculation of the coupling strength from high bias replicas matches that obtained from low bias replicas (see Supplementary Notes 3 and 7), showing explicitly that they have the same coupling strength.

We agree that this is a crucial point, so have added a more detailed and clear description to highlight that the coupling strength is the same for low and high bias replicas (page 5, left column, first paragraph).

7) At which value of V_T is Fig.4 recorded (please state in the Figure caption)? My guess is that it is at $V_T = -2.11$ V, since this is where Fig.3S refers to and where I know that $\alpha_0 = 2.5$ (why 2.6 ± 0.1 as stated in p.5, col.1, line 4?).

The new Figure 5 (corresponding to the old Fig. 4) is recorded at $V_{TG} = -0.8$ V and $V_T = -2.11$ V. The frequency of microwave irradiation was $f = 9.20$ GHz. We now explicitly state both values in the figure caption. The frequency $f = 9.20$ GHz corresponds to $\alpha_0 = 2.5$ from low-bias conductance replicas (see Supplementary Note 3). The coupling strength for high-bias conductance replicas, which is obtained independently, gives $\alpha_0 = 2.6 \pm 0.1$ at the same frequency.

We have updated the text to state a coupling strength of $\alpha_0 = 2.5$ for conductance replicas (page 5, right column, first paragraph).

8) In Fig.4e, how does this scatter plot compare to the power law formula used before? Perhaps show them together?

In the figure below, we show a comparison between the obtained values of α and the power law formula $\alpha = \alpha_0 \cdot 10^{P/20}$, for different values of coupling strength α_0 (colours).

The power law dependence does not capture the trend in the scatter plot, for any value of α_0 . We are not sure of the origin of the different power dependence. One possible explanation is that the semiconductor region of the junction has a finite length, meaning that local variations in the electromagnetic field might cause the AC voltage to be non-uniformly distributed across the interface of the junction leads. This is not accounted for by the assumption of the Tien-Gordon model, and the added complexity might give rise to a different power response of the circuit. A second possibility is that the microwave field suppresses superconductivity in the leads of the junction at large powers, due to enhanced quasiparticle populations in the superconducting film. This would lead to a decreased critical current and/or a suppressed switching current, not considered by the adiabatic model. At large powers, additional mechanisms are also known to cause deviations from the adiabatic model. In particular, a non-thermal ABS distribution and excitations to the quasiparticle continuum can lead to large discrepancies between the adiabatic theory and the full microscopic model [see new Ref. 52, Bergeret et al. Phys. Rev. B 84, 054504 (2011)]. Values for α obtained at large powers, where dips appear in the supercurrent, might therefore be unreliable.

We emphasise that the exact power dependence is not crucial to the conclusions we draw from our analysis: distortions to the supercurrent are well described by an adiabatic model at low microwave powers, and the estimated coupling strength is significantly lower than that obtained from conductance replicas. The difference in coupling strength is contradictory to the expectation for conductance replicas originating from Floquet-Andreev states, supporting our conclusions.

However, we find this discussion very interesting and decided to devote a new Supplementary Note to the power dependence of α (Supplementary Note 12, page 16). This Note discusses the device complexity, the spatial variation of the electromagnetic field and the additional distortions to the CPR from a non-thermal ABS distribution. The figure above is now Supplementary Fig. 22.

9) In supplement section 8, first line of third paragraph "...remarkable agreement with those at 0.8V", I suppose there should be a minus in front of 0.8?

Yes, that is correct. Thank you for pointing out this mistake, we have made the correction.

10) In supplement formula (S.3), should the "=0" be removed in the first argument of G on the right-hand side?

Yes, the formula was written incorrectly in the original Supplementary Material. The formulae (S.2) and (S.3) should read:

$$G_n \left(V_{MW}, V_{SD} + n \frac{hf}{e} \right) = G \left(V_{MW} = 0, V_{SD} + n \frac{hf}{e} \right) \left[J_n \left(\frac{eV_{MW}}{hf} \right) \right]^2 \quad (\text{S.2})$$

$$G(V_{MW}, V_{SD}) = \sum_{n=-N}^N G_n(V_{MW}, V_{SD}) \quad (\text{S.3})$$

We have implemented this change to Supplementary Note 4 (page 4).

Reviewer #2

[...] A previous publication, ref10, claims the observation of FAS without ever considering PAT, despite PAT being a generic phenomenon in all two-terminal electronic devices, and routinely presented in textbooks on superconductivity and quantum transport. The current manuscript is therefore extremely

timely, since it provides a much more detailed look at possible behaviours of Josephson junctions under radiation than reference 10, and provides a methodology for distinguishing between PAT of ABSs and FAS.

I can find no major error in analysis and find the experimental work of the highest standard. In particular, the data are consistently of superior quality compared to that of reference 10, likely due to their ability to keep their device cold while under radiation; sample heating seems to have been a big problem in reference 10.

However, I have several suggestions to the authors regarding the presentation of the data. I found the manuscript a little confusing and some of the data is challenging for a new reader to interpret in the current presentation. There seems to be two major issues: 1) the proposed scheme to discriminate between FAS and PAT is not clear to the reader until the discussion, even though this scheme could have been presented ‘as known’ from previous literature. 2) The first time the reader encounters power-dependent spectra in Fig 2, the data are very, very complicated, with signatures from the supercurrent, multiple ABSs, and multiple Andreev reflections (MARs). As the authors note, all these features produce replicas under applied microwave radiation, meaning that the data they present in Figure 2 is a mix of various different signatures, which is impossible for a first-time reader to decode. Even when the authors add guides, sometimes these get in the way; e.g. the green lines should be on the right side of the sub-figures in my opinion, since they risk obscuring important data. Conversely, I actually think the authors could have added guides in Fig 1e/f to immediately help the reader understand which features are which.

We thank the Referee for their kind comments on our work, and their suggestions for improving the clarity of our manuscript. We have made significant changes to the abstract, introduction, figures, discussion, and conclusions to improve the presentation of the data, following the recommendations of the Referee. We detail these changes in the following, and how they resolve the issues highlighted by the Referee.

1. The proposed scheme to discriminate between FAS and PAT is not clear to the reader until the discussion.

In the introduction to the paper, we now introduce coupling between superconductors and microwaves in a more general context. We describe the different mechanisms which can arise when photons couple to different parts of three-terminal devices, introducing FASs and PAT. We then consider the specific case of hybrid JJs, introducing the experiment of [new Ref. 44, Park et al. Nature 603, 421-426 (2022)] and the problem that conductance replicas with a Bessel function dependence and following a sum rule can come from both FASs and PAT.

Following this discussion, we further describe FASs and PAT, and the ways in which they differ. We highlight where they are expected to couple in the device, on which parameters they depend and the coherent/incoherent nature of the interaction. We comment on how these features might manifest in an experiment. During the discussion of the data, we refer back to these expectations, and how they distinguish the two effects. These changes address point 1 of the Referee.

2. The first time the reader encounters power-dependent spectra in Fig. 2, the data are very, very complicated.

We agree with the Referee that the data in the new Figs. 1, 2 and 3 are complicated. For this reason, we think that it is important to understand the conductance features of the device in the absence of microwave irradiation, before introducing effects related to microwaves. We do this in the new Figs. 1e and f, via the dependence on tunnel barrier voltage V_T and perpendicular magnetic field B_{\perp} . We use these measurements to distinguish between different conductance features, which we now explicitly label in the figures and describe in detail in the text.

Once we have established the origin of different conductance features, we present measurements under microwave irradiation. We have split up the original Fig. 2 into two figures: the new Fig. 2 corresponds to the tunnelling regime ($V_T = -2.11$ V), and the new Fig. 3 to the more open regime ($V_T = -2.08$ V). In this way, the reader is first introduced to conductance replicas of high bias features only, similar to those of [new Ref. 44, Park et al.]. The similarity is referred to explicitly in the text. Conductance replicas at low and high bias are easier to distinguish and can be commented on individually. In the open regime, we make the comparison of replicas in the supercurrent to Shapiro steps in the device, since supercurrent replicas are the clearest feature in this figure. The new Figs. 2 and 3 have also been extended to a larger vertical scale, and the labels have been moved to avoid conductance features. We believe that these changes more clearly describe the device behaviour in the absence of microwave irradiation, and gradually introduce the reader to the influence of microwave irradiation. These changes therefore address point 2 of the Referee.

To fix these issues, I would suggest first presenting data with no supercurrent, minimal MARs, and a single (or low number of) ABS, and show how this single ABS evolves under microwaves before presenting any of the figure 2 data. That is, the authors should show in Figure 1 that they can effectively reproduce the result of ref 10; I think most readers will be happy to see this early on. The data from Fig3c, or many of the sub-figures of supplementary figs S9, S10, S11 could serve. Once this ‘baseline’ is established, the data in figure 2 will be much easier to understand.

We propose a solution to the issues presented by the Referee in the answer above, and now comment on how they relate to the specific suggestions made by the Referee.

First, a crucial point of our work is the possibility to measure replicas in both the low-bias features (remnants of supercurrent) and high-bias features (sub-gap states), at the same tunnel gate voltage V_T . This is the result presented in the new Fig. 2, which allows us to compare the coupling strength of the two conductance features to the microwave irradiation. This is possible for the tunnel gate voltage set to $V_T = -2.11$ V. As identified above, the resulting spectrum contains many complicated features. We therefore believe that it is important to explain these features in detail, first in the absence of microwave irradiation (in the new Fig. 1), such that the reader can better understand the crucial results under microwave irradiation (the new Figs. 2 and 3).

Second, devices of the size used here cannot be tuned to a single ABS. The conductance spectrum at $V_{TG} = -1.4$ V, as in original Figs. 3c, S.9, S.10 and S.11, still contains complex features corresponding to many ABSs and resonances in the tunnelling probe. These data were also taken in a different regime of the device to the original Fig. 2, namely at a different top-gate voltage and a correspondingly different value of V_T . We believe that presenting these datasets before the data of the original Fig. 2 would add confusion, since the device was in a different regime and the plots cannot be directly compared with each other.

We agree with the Referee that it is important to establish a ‘baseline’ understanding of the device behaviour. For this purpose, we use the new Fig. 1 to clearly describe the conductance features in the device (using the changes outlined in the previous response). We also agree that it is important to have a clearer indication of the microwave response, with as little complexity as possible. We therefore split the original Fig. 2 into two, and first present data at $V_T = -2.11$ V. We make a direct comparison to the results of [new Ref. 44, Park et al.] and start with a simpler picture of conductance replicas emerging under microwave irradiation.

Here is a suggested flow:

- 1) As is for Fig1 a-e.
- 2) Fig 1f is replaced by data from Fig 3c/S9-11, i.e. ‘reproducing reference 10’. The current Fig 1f can be moved to the top of Figure 4, where the reader will want it anyway to compare with the rest of the sub-figures.
- 3) Use a paragraph to clearly outline the proposed delineation scheme.
- 4) Present the rest of the data basically as is, but now the authors will be able to reference everything back to the proposed discrimination scheme, which the reader will now already know about.

Based on the above discussion, we have implemented the comments in the following way:

1. The new Fig. 1 presents the data as before, but with clear labels on the origin of different conductance features. The new manuscript also features a largely extended introduction, where transport features are described in detail, to establish an understanding of the device behaviour in the absence of microwave irradiation. The features in Fig. 2 are then more readily understood.
2. The original Fig. 2 is split into two figures: the new Fig. 2, in the tunnelling regime ($V_T = -2.11$ V); and the new Fig. 3, in the more open regime ($V_T = -2.08$ V). The new Fig. 2 presents data ‘reproducing [Park et al.]’ as suggested by the Referee, in a gate voltage regime compatible with Fig. 1. This introduces the conductance replicas of low and high bias conductance features. The new Fig. 3 shows a regime where supercurrent replicas (reminiscent of Shapiro steps) are clearer, and the power dependence of low and high bias can be seen on the same plot.
3. We now use a paragraph before the discussion of the data to clearly outline experimental signatures which are different from FASs and PAT. We use existing knowledge and literature to inform this discussion, in a general context of microwaves interacting with superconducting devices.
4. During the discussion of the data, we refer back to the expectation for the different phenomena and state how the experimental result constitutes a delineation scheme for the different effects. In the conclusion, we explicitly list the delineation scheme, and how it relates to the expectations put forward in the introduction.

If the authors can rework the manuscript to make it a little more accessible to readers, it will be more than suitable for publication in Nature Communications. In fact, I don't see why it shouldn't be published alongside reference 10 in Nature itself.

We have made changes to the abstract, introduction, figures, discussion, and conclusions to make the text more accessible to readers, and to frame the results in a more general context of superconducting systems under microwave irradiation.

Minor comments:

- The references are out of order.

We thank the Referee for their comment, and have fixed the reference order in the updated manuscript.

- In Fig 1e, the features ‘bend’ at higher gate voltages. Is this simply due to not correcting the V_{SD} axis for the lowered device resistance? If so, I would suggest correcting for this, so readers understand that the gap, MARs etc should be energy independent.

The Referee is correct in stating that conductance features in the original Fig. 1e ‘bend’ due to the uncorrected V_{SD} axis. We have corrected this in the new Fig. 1 below, which is in the updated manuscript. All figures in the Main Text and Supplementary Material have been corrected to account for this mistake. We note that this does not change the conclusions of the paper, nor the results of data analysis. We thank the Referee for pointing out this issue in the data.

- Consider increasing the size of Fig 2, or split the data over two figures, so that the features are more clearly visible, especially on a printed page.

To aid the reader, we have made two key changes to the original Fig. 2:

1. We have increased the vertical size of plots, so that conductance features are more visible.
2. We have split the original Fig. 2 into two figures, new Figs. 2 and 3, each corresponding to a different regime of the tunnel barrier.

We agree with the Referee that this makes the data easier to read and understand. We present the high tunnel barrier regime first (originally Figs. 2f-j), such that the first result under microwave irradiation can be directly compared with previous work in the new Ref. 44 [Park et al.] (see comment above). We then present the more open tunnel barrier regime, to show the Shapiro steps more clearly in the device.

New Figure 2.

New Figure 3.

- When the authors are discriminating between features, there are basically five phenomena they could be: ABSs, MARS, the gap itself, supercurrent or Yu-Shiba Rusinov states (YSR not so likely given no other evidence of a quantum dot or spin impurity). Each of these presents either at different energy and/or has a different radiation power dependence. Perhaps this is worth stating explicitly, since I presume this is how the authors were able to distinguish between the features.

As pointed out by the Referee, we used the energy of conductance features, and their dependence on tunnel gate voltage V_T and perpendicular magnetic field B_{\perp} , to distinguish different phenomena. We highlight the key signatures here, and refer to an updated Fig. 1 of the Main Text to identify each feature:

- Supercurrent: conductance at $V_{SD} = 0$, most prominent for more positive V_T . Labelled by the green arrow in Fig. 1e, and at zero bias in Fig. 1f.
- Superconducting gap: conductance peak at high bias, most clearly visible at very negative V_T . Independent of B_{\perp} . Labelled by the white arrow in Fig. 1e.
- Andreev bound states: conductance features at bias values below the superconducting gap, which depend on B_{\perp} with a period of $200 \mu T$ (corresponding to the area of the superconducting loop). Labelled by the red dashed line in Fig. 1f, which depends on B_{\perp} .

- Multiple Andreev reflections: conductance features at more positive V_T which are regularly spaced in bias by $V_{SD} = 2\Delta/ne$. Labelled by blue arrow in Fig. 1e, and by the blue dashed lines in Fig. 1f.
- Resonances and disorder in the tunnelling probe: some conductance features which are independent of B_{\perp} and inside the superconducting gap suggest that they originate from resonances in the tunnelling probe. Conductance features can also couple to these resonances, giving a mirage effect of replicating some features attributed to ABSs [see new Ref. 50, Su et al. Phys. Rev. Lett. 121, 127705 (2018)]. Labelled by the red dashed lines in Fig. 1f.

Yu-Shiba Rusinov states are not present in this system, since there is no quantum dot or magnetic impurity.

In addition to the updated figure, we have expanded the discussion of Figs. 1e,f to clearly state how we distinguish different features in the conductance spectrum. We now establish a more thorough understanding of the device in the absence of microwave irradiation, which we hope will aid the reader in understanding the complexity of conductance features under microwave irradiation (see comment above).

In a model for PAT, all conductance features (supercurrent, MAR, ABSs, gap) are expected to have the same coupling to the microwave field, since photon interaction occurs in the tunnelling barrier for all charges transported across the barrier. The voltage spacing of replicas is hf/q , where q is the charge transported across the barrier for that process. In contrast, FASs constitute a change in the energy spectrum in the device, meaning replicas in the conductance features associated with ABSs only. We now state this explicitly in the discussion.

And finally, out of interest: Have the authors raised the issues with the editors of Nature and/or the authors of reference 10? Since PAT in Josephson junctions has been studied for many decades now (as the second last author here well knows!), it's not feasible for reference 10 to remain published as-is. They really must reanalyse their data to find the extent to which it can be explained by PAT.

Yes, we raised the issue both with the authors of Ref. 44 [Park et al. Nature 603, 421-426 (2022)] and the editors of Nature. On recommendation of editors of Nature Communications, we are investigating the possibility to submit a Matters Arising to Nature regarding the importance to consider PAT in interpreting conductance replicas arising under microwave irradiation.

Reviewers' Comments:

Reviewer #1:

Remarks to the Author:

In short, the authors have made good changes to improve the manuscript, and I am happy to recommend publication in its present form.

Reviewer #2:

Remarks to the Author:

The authors have addressed all my comments. The manuscript now reads very well, and is in my opinion ready for publication. I commend the authors for their thorough research and excellent work, which many others will no doubt find of value.

**Response to Reviewer Report of Nature Communications manuscript NCOMMS-23-25960-A
“Microwave-induced conductance replicas in hybrid Josephson junctions without Floquet–
Andreev states” by D. Z. Haxell *et al.***

Reviewer #1:

In short, the authors have made good changes to improve the manuscript, and I am happy to recommend publication in its present form.

Reviewer #2:

The authors have addressed all my comments. The manuscript now reads very well, and is in my opinion ready for publication. I commend the authors for their thorough research and excellent work, which many others will no doubt find of value.

We thank both Reviewers for appreciating the changes made to the manuscript and for recommending it for publication.